

# Upgrade and automation of the JPL Table Mountain Facility tropospheric ozone lidar (TMTOL) for near-ground ozone profiling and satellite validation

Fernando Chouza[1], Thierry Leblanc[1], Mark Brewer[1], and Patrick Wang[1]

[1]Jet Propulsion Laboratory, California Institute of Technology, Wrightwood, CA, USA

**Correspondence:** Fernando Chouza (keil@jpl.nasa.gov)

**Abstract.** As part of the international efforts to monitor air quality, several satellite missions were deployed (e.g. TROPOMI) and others, like TEMPO, are planned for the near future. In support of the validation of these missions, major upgrades to the tropospheric ozone lidar located at the Jet Propulsion Laboratory Table Mountain Facility (TMF) were recently performed. These modifications include the full automation of the system, which now allows unattended measurements during frequent
satellite overpasses, and a new receiver that extends the measurement capabilities of the system down to $100\,\mathrm{m}$ above surface.

The automation led to the systematic operation of the lidar during daily TROPOMI overpasses, providing more than 139 reference profiles since January 2018. Ozone profiles retrieved using the new lidar receiver were compared to ozonesonde profiles obtained from a collocated tethered balloon. An agreement of 5 % or better with the ozonesonde down to an altitude range of 100-m above ground was demonstrated. Furthermore, the stability of the receiver configuration was investigated.
Comparisons between the lowest point retrieved by the lidar and a co-located surface ozone photometer showed no sign of drift over a two-month test period. Finally, measurements from a 24-hour intensive measurement period during a stratospheric intrusion event showed good agreement with two free flying ozonesondes. These comparisons revealed localized differences between sonde and lidar, possibly owing to the differing vertical resolutions (about $52\,\mathrm{m}$ for lidar and at least $120\,\mathrm{m}$ for sonde).

# 1 Introduction

Ozone plays different roles in the troposphere depending on its location. At ground level, a high ozone concentration affects the air quality, posing a hazard for human health (WHO, 2003), animals and vegetation (Mauzerall and Wang, 2001), while in the upper troposphere, ozone acts as an effective greenhouse gas (Stocker, 2014). Tropospheric ozone has two major sources, namely stratospheric ozone downward mixing (Leblanc et al., 2011; Langford et al., 2018) and photochemical processes
involving carbon monoxide and volatile organic compounds in the presence of nitrogen oxides (Su et al., 2017). Although both these processes occur naturally, the increasing anthropogenic emission of ozone precursors as a result of the expanding industrial activity led to an increase in the tropospheric background ozone concentrations with respect to pre-industrial levels





(Horowitz, 2006; Young et al., 2013). The concentration of tropospheric ozone can fluctuate over relative small temporal and spatial scales as the result of different factors, including the emission rate of precursors, solar radiation intensity and advection processes (Hu et al., 2012). Although significant progress has been made during the last decades in the understanding and modeling of these processes and their relative impact, there are still major gaps in our knowledge. Additionally, and as a result

of the stricter emissions control policies and air quality regulations, there is an increasing need of high temporal and spatial resolution ozone concentration measurements for air quality analysis and forecasting (Moltchanov et al., 2015).

Over the past decade, several missions focused on the investigation and monitoring of atmospheric pollutants based on solar-synchronous satellites instruments (Veefkind et al., 2012; Levelt et al., 2018) and others, like the geosynchronous satellite TEMPO, are expected to be launched soon (Zoogman et al., 2017). While this kind of missions provide large spatial coverage

and important information about long-range transport processes, their coarse vertical resolution and daytime-only temporal coverage remain a limiting factor for addressing some key aspects of the pollutant life-cycle in the boundary layer and episodic events.

Due to its relative high temporal and vertical resolution, the lidar technique is able to provide measurements to close the gap between localized high temporal resolution measurements typically provided by in-situ instruments and the coarse resolution

measurements provided by satellites. This allows not only to study boundary layer processes, but also provides valuable information for satellite validation purposes (Newchurch et al., 2016). While this make lidars a very useful tool for atmospheric and validation studies, their high complexity typically requires the presence of a trained operator, posing a limitation to operation schedules and time-coverage. In addition the lidar technique inherently cannot provide reliable measurements at very near-range (i.e., typically below a few hundred of meters).

Hence, over the past few years, several efforts were conducted to increase the reliability of lidars, to automate their operation, and to extend downward their measurement range. As a result, several instruments capable of long-term unattended operation were developed (e.g. Engelmann et al., 2016; Strawbridge et al., 2018), and different approaches for low range measurements were proposed, including airborne measurements (Langford et al., 2011; Aggarwal et al., 2018), scanning lidars (Machol et al., 2009), and multi-receiver systems (Kuang et al., 2013; Farris et al., 2018).

The present work describes a series of upgrades to the JPL Table Mountain Facility tropospheric ozone lidar (TMTOL) hardware and software that allow the system to better reach the needs of current and future air quality and satellite validation studies. The upgrades include the possibility of conducting unattended measurements based on a given schedule and the extension of the measurement range down to 100 m above ground level (AGL).

The paper is organized as follows. Section 2 provides a brief description of the system previous to the modifications presented

in this paper. Section 3 describes the hardware and software modifications introduced as part of the system automation. Section 4 provides a description of the new receiver for very-near-range measurements. Section 5 presents the validation of the new receiver by means of a comparison with an ozone sensor deployed on a tethered balloon as well as a case study that shows the potential of unattended operations enabled by the system automation. A summary of the key achievements presented on this paper and an outlook of future developments are presented in section 6.





## 2 Instrument description

The TMTOL operations (34.3820° N; 117.6818° W, 2285 m above sea level (ASL)) originally started in 1991 with the alternate measurement of aerosols and ozone (McDermid et al., 1991). The system was redesigned in 1999 to provide routine measurements of tropospheric ozone in the middle and upper troposphere for the Network for the Detection of Atmospheric

Composition Change (NDACC) (McDermid et al., 2002). As part of the re-design, the aerosol measurement capability was removed and a new receiver, based on a larger Newtonian telescope (0.91 m diameter) and interference filters, replaced the spectrometer-based previous configuration.

Since the last description of the instrument presented in McDermid et al. (2002), a few modifications were introduced. The old Nd:YAG laser, which operated at a repetition rate of 10 Hz and a 266 nm pulse power of about 50 mJ, was replaced by

a Spectra Physics PIV-400, which contains two independent Nd:YAG lasers operating at a repetition rate of 30 Hz. Each side of the laser is followed by second- and fourth-harmonic generators, giving place to two independent 266 nm beams with a pulse power of about 65 mJ each. The fundamental and second-harmonic wavelengths are separated from the 266 nm beams by dichroic beamsplitters and redirected to beam dumps. Each of these two 266 nm beams is then focused into 3.6 m long cells (original cells were only 2 m long) made of 19-mm-outer-diameter stainless steel tubes ended by sapphire windows and filled

with high pressure gas for stimulated Raman shifting (SRS) to longer wavelengths (Haner and McDermid, 1990). One of these Raman cells is filled with $D_2$ at 4000 kPa, which shifts the 266 nm pump beam to 288.9 nm (first Stokes). The other cell is filled with $H_2$ at the same pressure, which shifts the pump beam to 299.1 nm (first Stokes). After passing through the Raman cells, both beams are re-collimated, expanded five times and sent into the atmosphere. The last turning mirrors are mounted on motor-controlled mirror holders, which allow the independent pointing of the beams for alignment purposes.

The lidar receiver remained almost unchanged with respect to the last system update. Up to now, the system operated with two receiver units to accommodate the dynamic range of the upper and lower troposphere atmospheric backscatter. The high altitude receiver, covering approximately the range between 6000 m and 17 000 m ASL, consists of a 0.91-m-diameter parabolic mirror with a focal length of 2.54 m. On its focal plane, an arrangement of two optic fibers collects the backscattered light and directs it to a filter and detector arrangement. This dual fiber arrangement allows the receiver to have two separate fields of view

(FOVs), and thus, by pointing each transmitted beam to a separate atmospheric volume, separate the two DIAL wavelengths. The low altitude receiver, covering a range between 3200 m and 6000 m ASL, consists of two independent refractive telescope arrangements fiber-coupled to the corresponding filters and detectors. The original setup included a chopper wheel which was removed due to thermal issues. Tests performed after this change showed no impact in the retrieved ozone profiles.

## 3 Automation

As mentioned before, one of the main objectives of the system upgrade was to add autonomous measurement capability, which would allow to perform scheduled measurements, as required for satellite missions validation, without adding workload on the lidar operators. Furthermore, the automation of this lidar will serve as a test bench for the future development of a fully autonomous mobile lidar system.



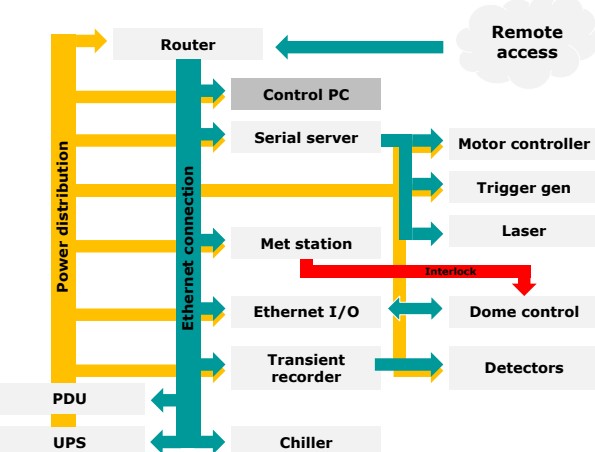

**Figure 1.** Block diagram describing the the power distribution (orange) and data transfer (green) between different subsystems of the lidar. Hatch hardware interlock based on rain sensing is shown in red.

Figure 1 presents a block diagram of the new automated system. The power distribution on the system is based on a $3\,\mathrm{kV\,A}$ APC UPS followed by an APC Ethernet controlled switched rack power distribution unit. This setup provides uninterrupted power and allows the remote power control and restart of all lidar subsystems, excluding the laser and the chiller unit.

The system control, data acquisition and storage is based on a standard desktop PC, which connects with the rest of the

devices with an Ethernet interface. For the case of existing subsystems that only support an RS232 interface (laser, alignment motor controller and the laser trigger generator), a Moxa NPort 5610-8-DT Ethernet-RS232 converter was included. The control and status acquisition of the dome and telescope hatch is implemented with a Moxa ioLogik E1214 remote Ethernet I/O device. Additional system monitoring is available through two Amcrest 2K webcams with Ethernet interface. These two cameras allow the operator to have a full view of the system, including the dome hatch and the laser. A meteorological station, a

Vaisala DRD11A rain sensor and an all-sky camera are connected to the control computer in order to provide real time weather information to the lidar control software. Since the system dome hatch does not have any protective window, an additional direct connection between the rain sensor and the dome hatch control was added in order to ensure that, even in the case of a crash in the control software or computer, the system cover will be closed in case of precipitation events.

The data acquisition system is based on Licel transient recorders. On its original configuration, four TR20-12 modules were

used to acquire the signals corresponding to the low and high range receivers. For the new near-range receiver (Sec. 4), a second rack with two TR20-16 (photon-counting and 16-bit analog detection) was added. Additionally, in order to simplify the automation process, the original Licel transient recorder rack interface (DIO-32HS) was replaced with an Ethernet card interface.

Laser and data acquisition timing are generated by a Stanford Research DG535 followed by a Quantum Composer Model

9520.



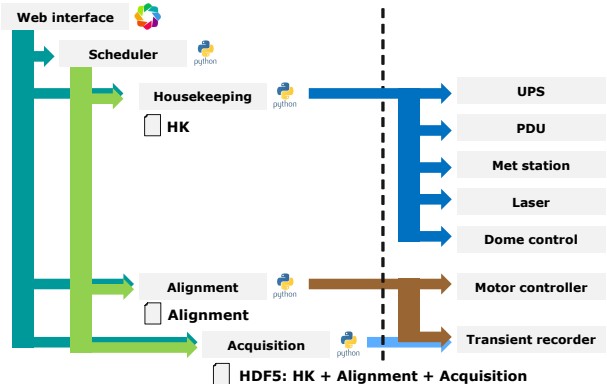

**Figure 2.** Block diagram of the new acquisition and lidar control software.

The new lidar control and data acquisition software were designed in a modular approach to ease the debugging process and reduce the complexity of porting the software to other lidar systems existing at TMF. Since the system is now designed to operate in autonomous mode, the possibility of analyzing the system status from a remote place is a very important feature. For this purpose, a web interface was developed. In this way, any device with a web browser connected to the TMF network

can be used to check the system status. The software described in this section was implemented in Python (www.python.org). The Python library Bokeh (http://bokeh.pydata.org) was used for the implementation of the system web interface and data acquisition visualization.

The general structure of the software is presented in Fig. 2. The interaction with the lidar hardware is implemented in the housekeeping, alignment and acquisition modules. The housekeeping module is in charge of performing most of the tasks that

were previously conducted by the lidar operator. This includes the monitoring of the power quality, turning on and off each lidar subsystem and analyzing if the meteorological conditions reported by the meteorological station are adequate for the lidar operation. The alignment module was implemented to control the alignment of the beams in order to optimize the data quality of the acquisition. Finally, the acquisition module is in charge of retrieving the data from the Licel transient recorders and storing the data.

For the data storage, the HDF5 format was chosen. The acquisition software is currently setup to store one file every minute. This includes the average of the acquired lidar profiles, as well as different environmental variables like the site meteorological conditions, last system alignment profiles and trigger timing information. In this way, a complete track of the system status is stored together with the actual lidar data. This information would help keep track of the system health and track down the cause of any issue found during the data analysis process.

The software and hardware automation described in this section has been operational since beginning of 2018. Together with the regular 2-hour experiments typically conducted at TMF as part of NDACC activities, regular daytime measurements were carried out in support of the validation of the TROPOMI instrument onboard the S5P satellite. In both cases, experiments were





**Table 1.** Number of experiments conducted during 2018 as part of the TROPOMI validation and NDACC activities. Measurements reported for September are only up to September 13.

| Month | TROPOMI | NDACC |
|-------|---------|-------|
| January | 16 | 9 |
| February | 13 | 10 |
| March | 9 | 12 |
| April | 17 | 16 |
| May | 26 | 20 |
| June | 17 | 12 |
| July | 12 | 13 |
| August | 20 | 16 |
| September | 9 | 5 |

conducted in autonomous mode, independently if an operator was on-site or not. Table 1 provides an overview of the number of experiments conducted after TMTOL automation upgrade.

## 4 Near-range receivers

As part of the lidar upgrade, a new receiver for the retrieval of ozone profiles between $100\,\mathrm{m}$ and $1000\,\mathrm{m}$ AGL was built. Since this altitude range typically overlaps with the atmospheric boundary layer (ABL) and given that the typically high aerosol concentration found in this layer can affect the accuracy of the DIAL technique, a quantification of this effect and an evaluation of alternative solutions to mitigate it were conducted.

Equation 1 presents the different factors to be considered during the ozone retrieval process (Leblanc et al., 2016b)

$$N_{O_3}(z) = \frac{1}{\Delta\sigma_{O_3}}\left[\frac{\partial}{\partial z}\left(\ln\frac{P_{OFF}(z)}{P_{ON}(z)}\right) - \Delta\sigma_M N_a(z) - \left(\sum_{ig}\Delta\sigma_{ig}(z)N_{ig}(z)\right) - \Delta\alpha_p(z) + \Lambda\beta(z) + \Lambda\eta(z) + \Lambda Z(z)\right] \quad (1)$$

where $N_{O_3}$ is the retrieved ozone number concentration, $\Delta\sigma_{O_3}$ is the differential absorption cross-section, $P_{OFF}$ and $P_{ON}$ are the number of photons collected by each detector after pile-up correction and background subtraction, $\Delta\sigma_M$ is the differential Rayleigh cross-section along the beam path up to altitude z and back, $N_a$ is the air number density, $\Delta\sigma_{ig}$ is the differential cross-section along the beam path up to altitude $z$ and back for the atmospheric constituent $ig$, $N_{ig}$ is the number concentration of the constituent $ig$, $\Delta\alpha_p$ is the extinction differential due to particles and computed along the beam path up to altitude $z$ and back, $\Lambda\beta$ is the effect of the difference in the atmospheric backscatter coefficient between both DIAL wavelengths, $\Lambda\eta$ represents the effect of the difference in efficiency of both receivers and $\Lambda Z$ represents the effect of timing differences between both DIAL channels.





Proposed solutions for the correction of the aerosol effects include the dual DIAL (Kovalev and Bristow, 1996) and the Raman DIAL (McGee et al., 1993) techniques, as well as aerosol corrections based on the assumption of aerosol properties (e.g. Eisele and Trickl, 2005). In the first case, three different wavelengths to create two DIAL pairs are needed. This could be achieved by using remaining pump, first and second Stokes wavelengths coming out from one of the two Raman cells.

Although this might be an attractive idea, this would require a reoptimization the Raman cells to increase the amount of pump power converted to second Stokes. This reoptimization would require lowering the cell pressure (Haner and McDermid, 1990) and will have as secondary effect the reduction on the first Stokes conversion efficiency. This will, in turn, reduce the accuracy of the mid and long range channels. As an alternative, the two DIAL wavelengths and the remaining pump of one of the cells could be used. In this case, since two different beams would have to be received, the overlap function of the channels would vary

independently after system alignment, making difficult to achieve unbiased measurements at very short ranges. Additionally, it has to be mentioned that although dual DIAL measurements help to reduce the aerosol influence on the retrieval, it is at the expense of increasing the uncertainty on the ozone retrieval (Kovalev and Bristow, 1996; Wang et al., 1997; Alvarez et al., 2011)

On the other hand, the Raman DIAL technique, as in the case of dedicated aerosol Raman lidars, make use of the atmospheric

Raman scattering by nitrogen or oxygen to retrieve aerosol extinction profiles. Based on this, a correction for the extinction and backscatter differential between DIAL wavelengths can be calculated. Because the daytime operation is important for the validation satellite-borne ozone instruments and given that the Raman scattering from atmospheric oxygen and nitrogen is more than two orders of magnitude smaller than the Rayleigh scattering, the use of wavelengths in the blind region of the solar spectrum is preferred. This discards the possibility of using Raman scattering stimulated by the standard DIAL wavelengths

used by TMTOL (288.9 nm and 299.1 nm), and limits the application of the Raman DIAL technique to the vibrational Raman scattering of nitrogen (283.6 nm) and oxygen (277.5 nm) induced by the non-converted 266 nm energy leaving the Raman cells.

The measurements performed at the output of both Raman cells showed a remaining 266 nm pulse power of about 2 mJ on each side. Considering that the system reported by Lazzarotto et al. (2001) used a 266 nm laser with a pulse power of

120 mJ, a repetition rate of 10 Hz and a 20-cm-diameter telescope to provide 5 percent accuracy ozone profiles up to 700 m, a quick estimation indicates that the available 266 nm power would not meet the range requirements previously mentioned while keeping a reasonable size receiver and similar accuracy and averaging time as the previously mentioned study. Although using the already existing 0.91-m-diameter telescope receiver would have increased by a factor of 20 the received power, achieving full overlap at altitudes as low as 100 m would have required major modifications on the system.

As an alternative to the previously mentioned approaches, a standard DIAL receiver based on the 266/288.9 nm wavelength pair and a small 50-mm-diameter refractive telescope was implemented. Although the use of the 266 nm laser wavelength is typically discourage for ozone DIALs due to the very high ozone absorption at this wavelength, this doesn't pose a limitation in this case, where only a short range is covered. Among available wavelength pairs (Fig. 3), the 266/288.9 nm pair is the one that maximizes the $\Delta\sigma_{O_3}/\Delta\lambda$ ratio. The high absorption represents an advantage as it improves the accuracy of the DIAL retrieval

for a given signal-to-noise ratio (SNR), reduces the impact of aerosols on the retrieval and the use of the output corresponding to





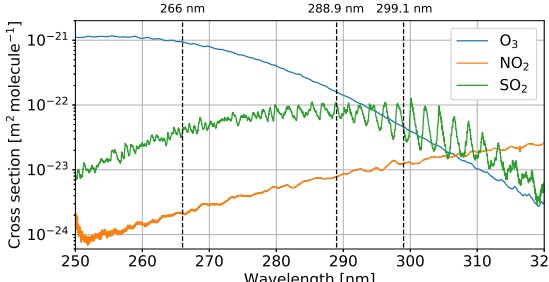

**Figure 3.** Absorption cross section for ozone (blue), nitrogen dioxide (green) and sulfur dioxide (orange) in the UV region of the spectrum (300 K; 775 hPa). Wavelengths transmitted by TMTOL are also shown (back, dashed)

only one Raman cell makes the alignment process easier as it helps to reduce the overlap difference between the two channels. Furthermore, this wavelength pair minimizes the cross-sensitivity with $SO_2$ compared with the other available pairs, and has a smaller cross-sensitivity with $NO_2$ than the 288.9/299.1 nm pair.

Figure 4 presents a comparison of the error introduced in the ozone retrieval as a function of the selected wavelength pair and
aerosol Ångström exponent for a given aerosol distribution. The aerosol distribution corresponds to an optical depth (OD) of 0.1 assuming a lidar ratio of 70 sr. Given that the total-column OD rarely goes above 0.1 at TMF (based on AERONET data) and considering that the proposed aerosol distribution contains a strong gradient between 400 m and 500 m, the results obtained from this estimations can be considered a worst case scenario for typical operation conditions. The results are compatible with the analysis based on the $\Delta\sigma_{O_3}/\Delta\lambda$ ratio. The largest rejection to aerosol influence is obtained by the 266/288.9 nm
wavelength pair, with a maximum deviation of less than 20 percent for an Ångström exponent equal to 0 and a typical ozone concentration of $1 \times 10^{18}$ m$^{-3}$. In the case of a more typical Ångström exponent of 1, the deviation is reduced to 10 percent, assuming the same ozone concentration.

Since only short ranges are covered by this receiver (Fig. 5), a 45:55 pellicle beamsplitter (Thorlabs BP145B5) is used to divide the received power and send it to each of the receivers. Although this means losing half the received power, the use of a
pellicle beamsplitter reduces the difference in the optical path of each receiver compared to a traditional thick beamsplitter, and thus, minimizes the difference in the overlap function between receivers. The 288.9 nm receiver arm uses a 2-inch 288.9 nm band-pass filter (1 nm full width at half maximum (FWHM)) to block solar light and the 266 nm DIAL wavelength. The 266 nm receiver uses a stack of two off-the-shelf 1-inch band-pass filters, a 10 nm FWHM filter that blocks most of the incoming solar radiation followed by a 1 nm FWHM that increases the rejection of the solar background and the 288.9 nm wavelength. After
filtering, the beams are focused over the surface of H11901P-113 Hamamatsu photomultiplier tubes (PMTs), giving place to an image of about 2 mm diameter for far-field radiation. Because both cells output non-converted 266 nm laser radiation, the laser and acquisition triggering was modified to interleave the pulses of each of the DIAL wavelength pairs. This eliminates possible cross-talk between DIAL receivers.



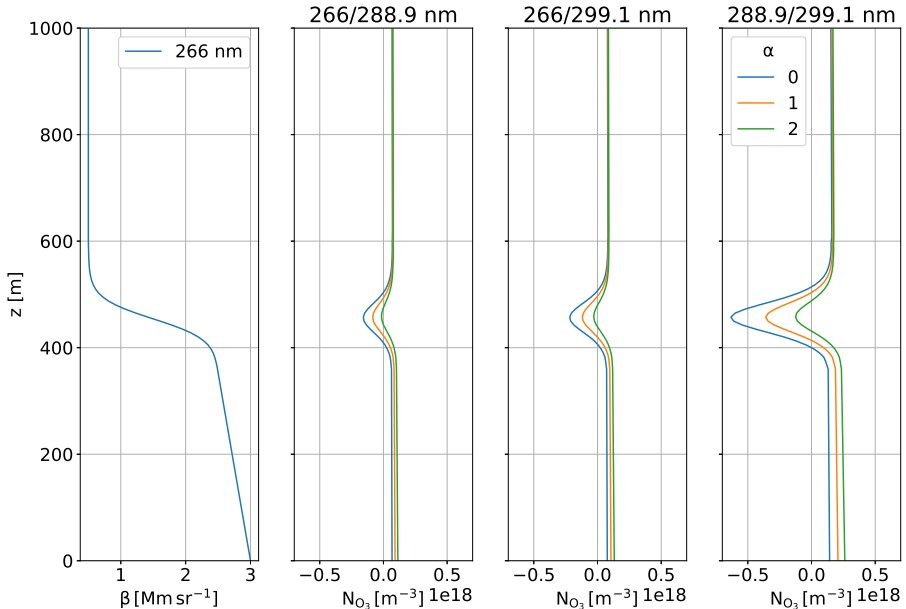

**Figure 4.** Aerosol error contribution (differential backscatter and extinction contributions) based for the different available DIAL wavelength pairs and three different Ångström exponents ($\alpha$).

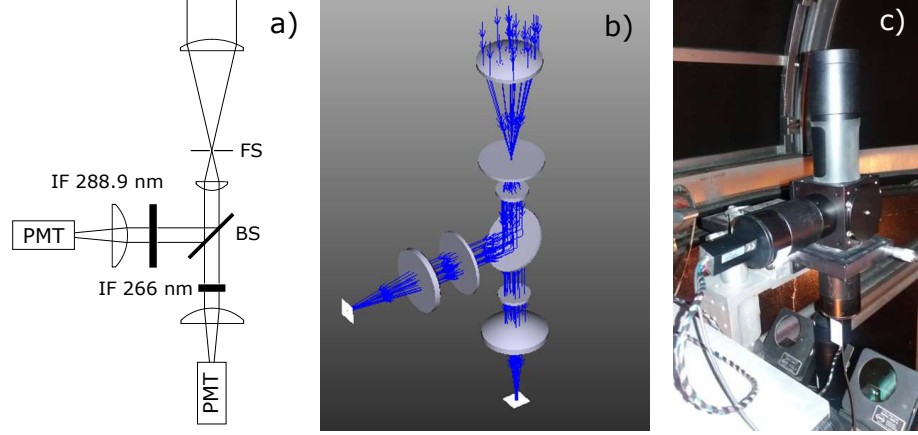

**Figure 5.** a) Scheme of the new very-near-range receivers. BS is a pellicle beam splitter, IF are interference filters and PMT are photomultipliers. b) Photo of the actual setup. Motor-controlled mirrors corresponding to the main wavelength pair of the system (288.9/299.1 nm) are shown on the bottom.





The signals generated by the PMTs are fed into two Licel TR20-16 transient recorders modules (photon-counting and 16-bit analog detection). Because measurements between $100\,\text{m}$ and $1000\,\text{m}$ AGL require a receiver with a dynamic range of about $30\,\text{dB}$, the combination of analog and photon counting detection allows to obtain a good compromise between accuracy and averaging time. First tests conducted with the receiver setup previously described showed strong oscillations in the first bins of the analog detection channels that compromised their use. In order to investigate the causes for this issue, artificial signals with similar slew-rate ($750\,\text{kV}\,\text{s}^{-1}$) to the ones generated by the receiver were fed to the Licel transient recorders. The results showed the same oscillations, indicating the the problem is most likely associated with the anti-aliasing filter in conjunction with the high slew-rate signals generated by the laser pulses entering on the receiver field of view at very low altitudes. Based on these tests, the receiver alignment was modified to reduce the received signal slew-rate and thus, minimize the impact of this issue.

Along with the dynamic range consideration mentioned before, the retrieval of very-near-range measurements typically involve a careful optical design in order to achieve full overlap at low altitudes while minimizing the system field of view and associated solar background. Since any range dependent difference between the efficiency of each of both receiver arms will be directly translated into a source of error during the retrieval process, special care has to be taken with regard to differences in the optical paths, inhomogenities on the PMTs surface sensitivities, timing and linearity of the transient recorders.

As part of the tests conducted on the Licel transient recorders, a sine wave generated by a signal generator was fed into both transient recorders via a T and a set of matched length cables. Signals triggering the signal generator and each of the receivers were generated by a Quantum Composer 9250 pulse generator. The results show a relative trigger delay of $23\,\text{ns}$ between the transient recorders user for this very-near-range receiver. In order to discard differences in the signals used to trigger each transient recorder, the common trigger input available in rack was used to trigger both transient recorders from the same trigger source. The results were consistent with the ones observed when triggering both transient recorders independently. Tests conducted along several days showed no change in the relative delay. Although this delay represents only about half sampling bin, the DIAL technique is very sensitive to relative delays between on and off sides, especially at very short ranges. It can be shown that the effect of a delay between the trigger of both DIAL channels $\Lambda Z$ can be written as

$$\Lambda Z(z) = 2\frac{\partial}{\partial z}\left[\ln\left(1 + \frac{\Delta R}{z}\right)\right] \qquad (2)$$

where $\Delta R$ is the shift in meters of the on channel with respect to the off channel.

Figure 6 shows the error generated as a function of the range for a set of delays between the on and the off channels of the $266/288.9\,\text{nm}$ wavelength pair.

As can be seen, for the delay observed between the two transient recorders ($23\,\text{ns}$ or $3.45\,\text{m}$), an overestimation of about $0.5 \times 10^{18}\,\text{m}^{-3}$ is introduced. For a typical concentration of $1 \times 10^{18}\,\text{m}^{-3}$, this represents an overestimation in the ozone concentration of about 50 percent at $100\,\text{m}$. By introducing a relative delay between the trigger signals of both transient recorders, this error source was eliminated.





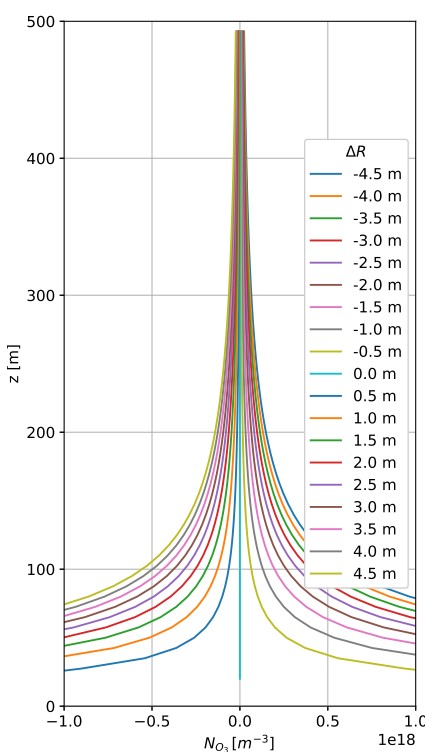

**Figure 6.** Error generated as a function of the delay $\Delta R$ between the 266 nm and the 288.9 nm channels. Positive delays correspond the the 266 nm channel being triggered after the 288.9 nm channel.

After correcting the effects of the relative trigger delay, and in order to further investigate the systematic error sources of the receiver, an estimation of the differential efficiency was performed. The differential efficiency condenses a set of different error sources, including differences in the receiver arm overlap functions, PMT surface sensitivities, PMT and transient recorder linearity, among others. The differential efficiency contribution to the ozone retrieval can be written as

$$\Lambda\eta(z) = \frac{\partial}{\partial z}\left(\ln\frac{\eta_{OFF}(z)}{\eta_{ON}(z)}\right) \tag{3}$$

where $\eta$ is the efficiency of each receiver arm.

In order to evaluate the actual contribution of differential efficiency $(\Lambda\eta)$ to the retrieval accuracy, a set of tests replacing the 266 nm interference filter by a second 1-inch 288.9 nm interference filter was conducted. Because the power of the transmitted 288.9 nm beam is about 10 times larger than the one of the 266 nm beam, an OD1 filter was added on the 266 nm arm in order



to achieve a signal with a similar intensity. Since the wavelength received by each receiver arm is the same in this configuration, it can be shown that

$$k_{inst} \frac{\eta_{OFF}(z)}{\eta_{ON}(z)} = \frac{P_{OFF}(z)}{P_{ON-288.9}(z)} \tag{4}$$

where $k_{inst}$ is a constant that condenses all range-independent instrumental parameters (i.e. transmitted power, optical efficiency), $P_{ON-288.9}$ is the power received on the 266 nm detector when the 266 nm IF filter is replaced by a 288.9 nm filter. Replacing Eq. 4 in Eq. 3

$$\Lambda \eta(z) = \frac{\partial}{\partial z} \left( \ln \frac{P_{OFF}(z)}{P_{ON-288.9}(z)} \right) \tag{5}$$

The result of the test conducted with this configuration is presented in Fig. 7. The profiles were retrieved based on a 20 minutes measurement during daytime. The derivative of the logarithm is calculated with a fourth-order Savitsky-Golay derivative filter and a window length of 9 samples for the analog detection mode and 31 samples for the photon-counting mode.

As can be seen, the differential efficiency contribution of the analog pair is very close to zero for altitudes above 50 m AGL, while for the case of photon-counting channels, no differential efficiency effects are observed above 200 m AGL. The difference observed between analog and photon-counting channels can be attributed to the non linearity of the photon-counting channels at high count-rates due to pile-up effect.

## 5 System validation

### 5.1 Validation with tethered-balloon

As part of the validation efforts of the new very-near-range channels, a set of measurements by an ozonesonde tethered system were carried out. This experiment was intended to provide a link between surface ozone measurements available at TMF (Thermo Scientific Model 49i) and the first valid data-points of the new receiver (about 100 m AGL). In this way, the minimum achievable range and accuracy of the new setup can be determined. The ozone measurement system consists of an iMet-1 radiosonde and an EN-SCI ECC ozonesonde tied to a balloon and driven by a winder/unwinder system built by NOAA (Schnell et al., 2016). This setup provides vertical profiles of pressure, temperature, humidity, GPS position and ozone concentration up to altitudes of about 200 m AGL. The profiles corresponding to one of the tests carried out as part of the validation process on June 7 are shown in Fig. 8. As can be seen, there is a very good agreement (less than 5 % difference) between the sonde and the lidar as well as between the sonde and the surface ozone measurements. Lidar retrievals look unbiased down to about 70 m AGL, with some deviations (less than 10 %) between 70 m AGL and 100 m AGL. These deviations might have different causes, like aerosol contamination or electronic noise.

The lidar measurements correspond to 1 hour average (19:58-20:58 UT), while the sonde measurements where conducted for about 30 minutes (19:45-20:15 UT). For the lidar retrieval between ground and 2600 m AGL, a 17-sample Savitsky-Golay



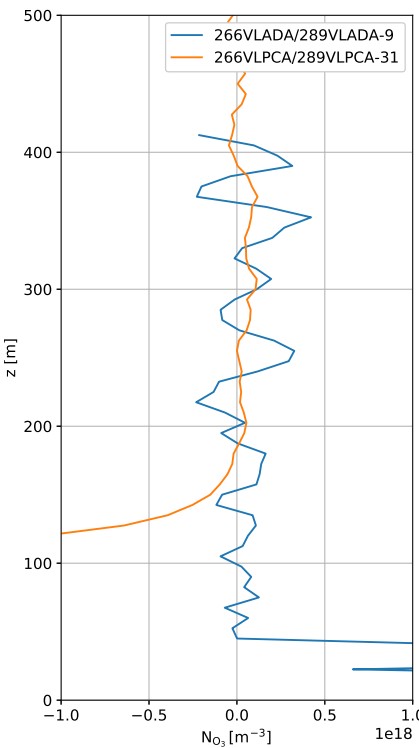

**Figure 7.** Error of the new very-near-range receiver as a result of the differential efficiency $\Lambda\eta(z)$ using photon-counting (solid, orange) and analog detection mode (solid, blue).

derivative filter was used, giving place to a $52\,\mathrm{m}$ vertical resolution following the definitions presented in Leblanc et al. (2016a). Above $2600\,\mathrm{m}$, the vertical resolution decreases to $200\,\mathrm{m}$. Uncertainties are provided according to the definitions presented in Leblanc et al. (2016b).

In the first $15\,\mathrm{m}$, the sonde measurements indicate an increasing ozone concentration with respect to the measurements conducted at surface level. This variation in the ozone concentration was observed repeatedly during the tethered-balloon validation process, suggesting that a comparison between the first valid lidar point and surface measurements might not always be a good validation approach. This result is compatible with previous studies conducted in forested areas (Gerosa et al., 2017; Makar et al., 2017).

While regular tethered balloon experiments would have provided further confidence on the receiver accuracy and minimum achievable range, tethered balloon operations demonstrated to be difficult at TMF due to relative scare low-wind conditions and the high amount of trees surrounding the site.





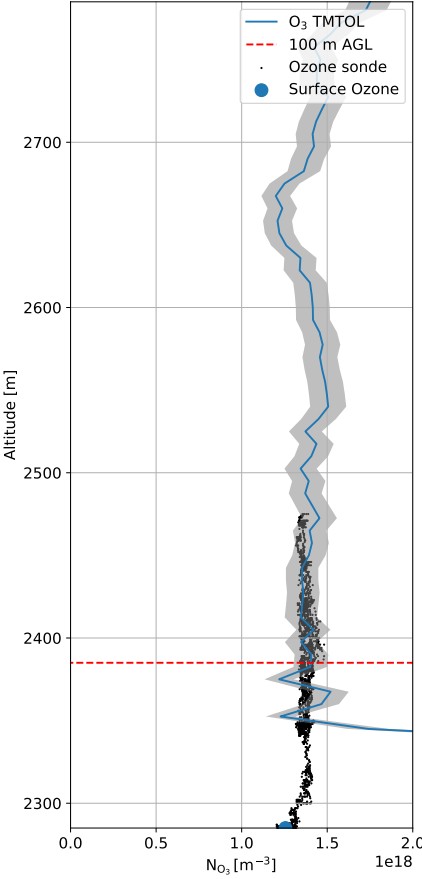

**Figure 8.** Comparison between the TMTOL very-near-range receiver retrieval (solid, blue), an ozonesonde tethered system (dots, black) and a surface ozone monitor (dot, blue). Lidar retrieval $1 - \sigma$ uncertainty is shown (gray, shaded).

## 5.2    May 25th: 24h case-study

In the frame of the system upgrade validation, a 24-hour run (25-05-2018 1:00 UT to 26-05-2018 1:00 UT) was conducted during a forecasted stratospheric intrusion. During the experiment, two sets of ozone EN-SCI ECC ozonesonde, i-Met 1 and Vaisala RS-41 radiosondes were launched to validate the retrievals of the very-near-range channels.

5      In order to provide a general overview of the synoptic meteorological situation during the stratospheric intrusion event analyzed in this section, MERRA-2 reanalysis ozone concentration, humidity and winds are shown in Fig 9 for two different model levels (775 hPa and 550 hPa) at 9:00 UT. The 775 hPa model level corresponds approximately to the TMF site surface, while the 550 hPa level corresponds approximately to an altitude of 5 km ASL. The general synoptic situation was dominated by a cyclonic system in the north and an anti-cyclone in the south. Ozone concentration at 775 hPa shows only slight spatial



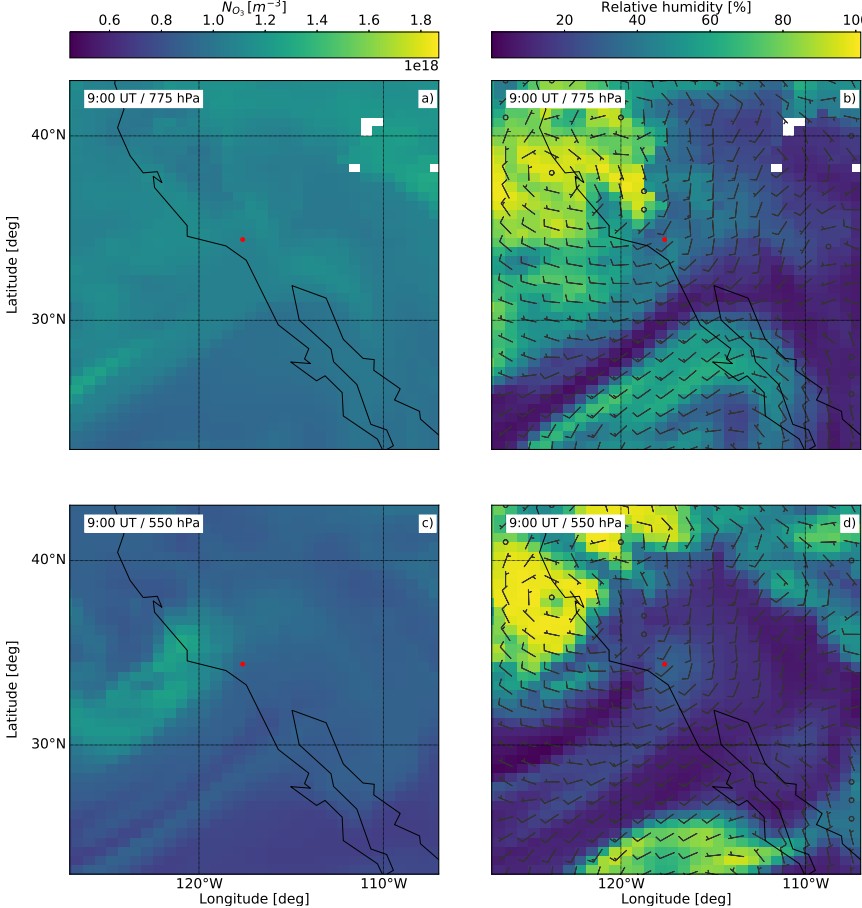

**Figure 9.** MERRA-2 reanalysis at 9:00 UT on 25 May 2018 for two different pressure levels (775 hPa and 550 hPa). a,c) Ozone number concentration. b,d) Relative humidity and wind speed/direction. TMF site location is indicated with a red dot.

variations around TMF, while at 550 hPa, an ozone-rich dry plume associated with the stratospheric intrusion can be seen approaching TMF from the west.

Retrieved ozone concentration profiles in the lower troposphere for this intensive measurement period are shown in Fig. 10a, together with a detailed view of the first 500 m AGL (Fig. 10b) and a comparison with a collocated surface ozone measurement unit (Fig. 10c). These retrievals correspond to the merging of the new very-near-range channels and the preexisting mid-range channels. Between 2355 m and 2600 m ASL, retrievals are based on the analog channels of the very-near-range receiver (52 m vertical resolution), while between 2550 m and 3185 m ASL are based on the photon-counting channels (200 m vertical resolution). Finally, above 3085 m and up to 6000 m ASL, the retrieval corresponds to the mid-range receivers (380 m vertical resolution). In the overlap region between the very-near-range and the mid-range receiver, the mean is presented. The temporal resolution is 30 minutes.





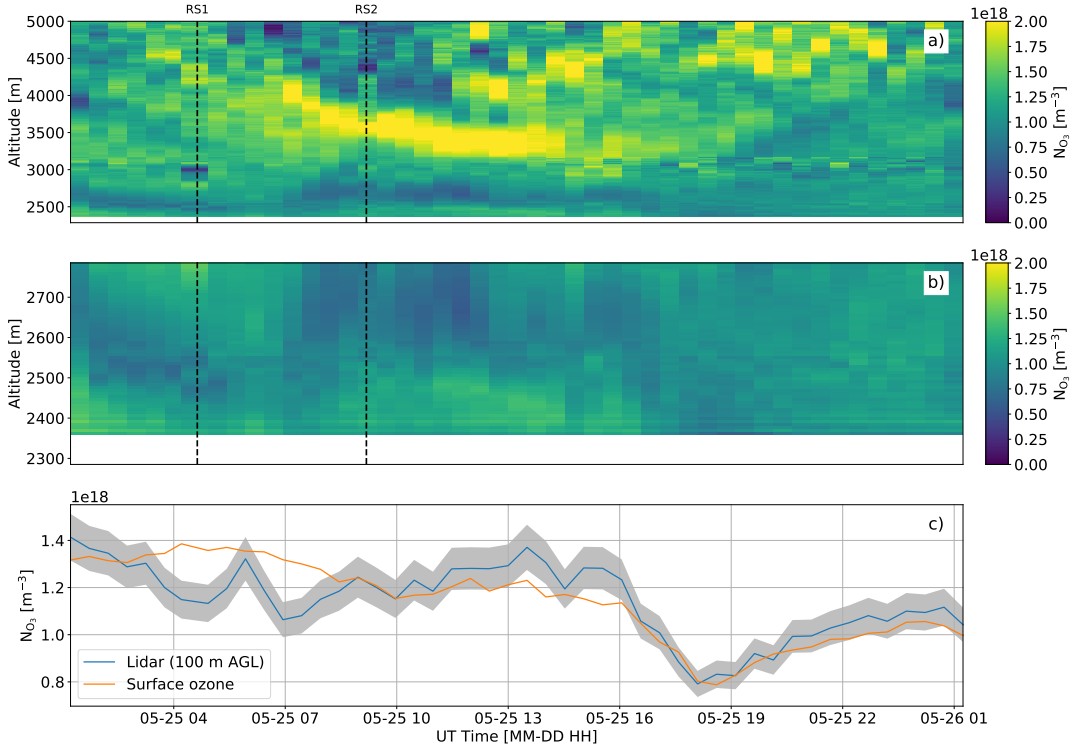

**Figure 10.** Experiment result overview. a) Ozone number concentration between ground level and $5000\,\mathrm{m}$. b) Detail of the first $500\,\mathrm{m}$ AGL acquired with the new very-near-range receiver. c) comparison between the surface ozone measurements and the lidar retrieval at $100\,\mathrm{m}$ AGL (together with $1-\sigma$ uncertainty).

A first qualitative comparison indicate a good agreement between the top of the very-near-channel and the bottom of the mid-range channels as well as between the $100\,\mathrm{m}$ AGL lidar retrieval and the surface ozone measurements. Although this doesn't provide a full validation of the new channel, it provides further confidence to the tests performed with the tethered balloon system and presented in the previous subsection.

5     The high ozone concentration visible in Fig. 10a above $3000\,\mathrm{m}$ is in qualitative agreement with a stratospheric intrusion event and with the forecast/reanalysis provided by the models. Below the intrusion, a thin layer of about $200\,\mathrm{m}$ depth characterized by a relatively low ozone concentration can also be recognized. Since this kind of fine structures are typically not reproduced by operational models and since they provide valuable information to evaluate the performance of the receiver, a comparison between the lidar retrieval and the two ozonesondes launched during the experiment is presented in Fig. 11. This will allow to

10    determine whether or not this corresponds to a real feature and what is the expected accuracy of the new receiver.





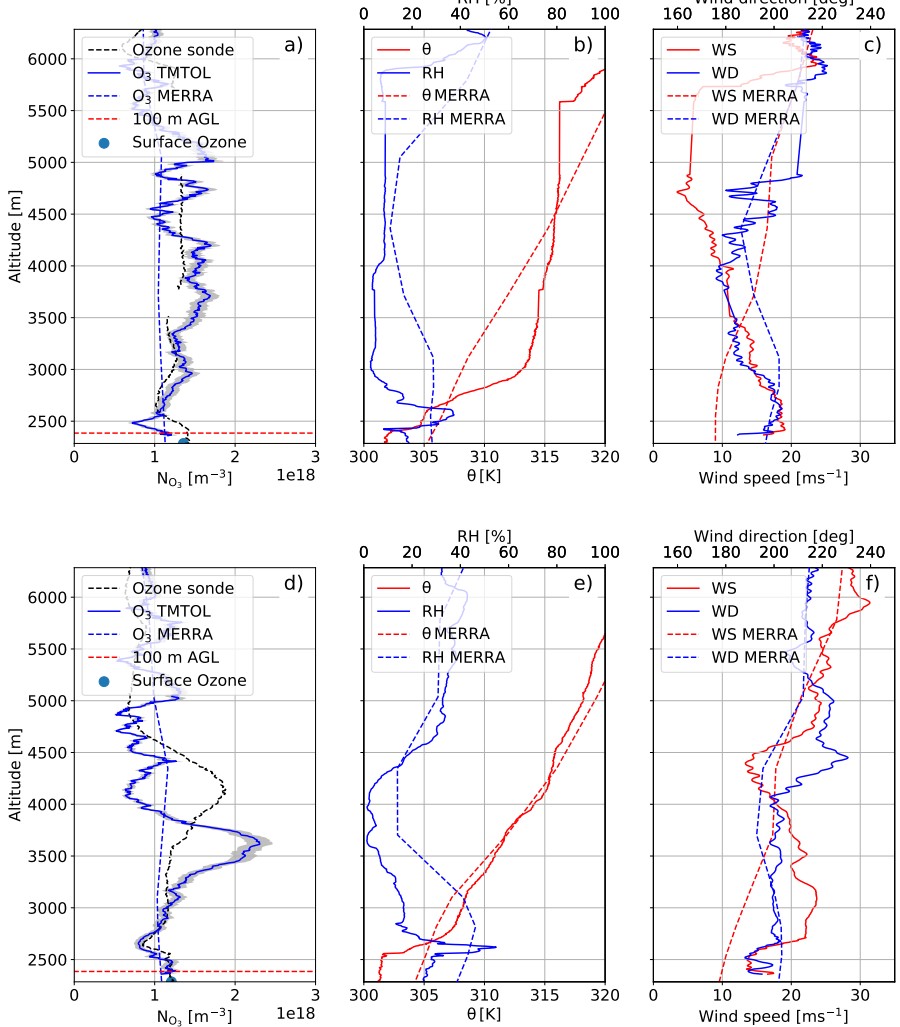

**Figure 11.** Comparison between the retrieved lidar ozone profile (together with $1 - \sigma$ uncertainty), collocated radiosondes, surface ozone meter and MERRA-2 reanalysis at the time of the first (upper row, 4:40 UT) and second launch (lower row, 9:10 UT). a,d) Lidar ozone profile (solid, blue), ozone sonde profile (dashed, black), MERRA-2 ozone concentration and surface ozone measurements (light blue). b,e) Radiosonde potential temperature (solid, red), MERRA-2 potential temperature (dashed, red), radiosonde relative humidity (solid, blue) and MERRA-2 relative humidity (dashed, blue). c,f) Radiosonde wind speed (solid, red), MERRA-2 wind speed (dashed, red), radiosonde wind direction (solid, blue) and MERRA-2 wind direction (dashed, blue).

Figure 11 presents the intercomparison between the two ozone sondes and the lidar-retrieved ozone profiles. Potential temperature, humidity and wind profiles retrieved by the radiosonde are also shown. During the launch of this first sonde (4:40 UT), no strong signature of the atmospheric intrusion was visible neither in the lidar nor in the ozone sonde. A general good agreement between the lidar- and the sonde-retrieved ozone profile is visible in Fig. 11a. In the very-near range, the thin layer





of low ozone concentration visible in Fig. 10 can be found between $100\,\text{m}$ and $200\,\text{m}$ AGL, while in the case of the sonde, a thicker and less pronounced layer of low ozone concentration was observed between $200\,\text{m}$ and $700\,\text{m}$ AGL. As can be seen in Fig. 11b, this layer sits on top of a well mixed layer that extends up to $120\,\text{m}$ AGL and is trapped between two inversions. The relative high humidity of this layer, of up to about 40%, suggests that it might be a stable marine air layer advected from the ocean over TMF.

In contrast to the previous case, the retrievals corresponding to the second launch (9:11 UT) show a strong signature of the stratospheric intrusion on both, the lidar and the sonde retrievals. While the lidar shows the intrusion maximum ozone concentration at $3.8\,\text{km}$ ASL, the sonde shows it at about $4.2\,\text{km}$ ASL. The magnitude of the ozone concentration is also different, with the sonde showing a peak value of about $1.8 \times 10^{18}\,\text{m}^{-3}$ and the lidar $2.3 \times 10^{18}\,\text{m}^{-3}$. This difference can be attributed to different causes, including ozone spatiotemporal variability combined with the sonde drift and differences between the vertical resolution of the sonde and the lidar. Compared to the previous case, the lidar shows a better agreement with the surface ozone measurement and the sonde in the first hundreds of meters. The thin low ozone concentration layer observed by the lidar and the first launched sonde was also observed during the second launch. Nevertheless, in this case, the agreement between the lidar and the sonde is much better regarding both, its vertical extension and ozone concentration. As in the previous case, this thin layer is trapped between two inversions and has a high relative humidity. HYSPLIT backward trajectories calculated for this layer (not shown) confirmed the marine origin of this layer.

### 5.3 Long-term stability

Since optical stability is a key factor in order to obtain reliable unbiased measurements in the very-near range, a long-term comparison between the first valid lidar data point ($100\,\text{m}$ AGL) and the collocated surface ozone meter deployed at TMF was conducted. The results, corresponding to a period of 2 months between 13 July and 12 September 2018, are presented in Fig. 12. A good correlation between these two quantities is observed, with consistent 10% lower values recorded by the surface ozone meter compared to the lidar. A similar difference was observed during the tethered balloon validation experiments (Fig. 8) and is mentioned in different works conducted on very-near ground zone vertical profiles as being related to ozone depletion in the first few meters above ground (Fontan et al., 1992). As additional validation, a similar comparison was conducted between the historical ozone sonde profile record from TMF and available collocated surface ozone measurements. The result of this comparison show a very similar difference of about 5-10% between ozone sonde measurements at $100\,\text{m}$ AGL and surface ozone measurements.

### 6 Conclusions

As part of the efforts conducted at TMF to provide accurate and regular measurements for satellite validation and process analysis purposes, a set of modifications were introduced in the TMTOL system. As a result, the system is now capable of performing measurements autonomously from about $70\,\text{m}$ AGL up to about $15\,\text{km}$.





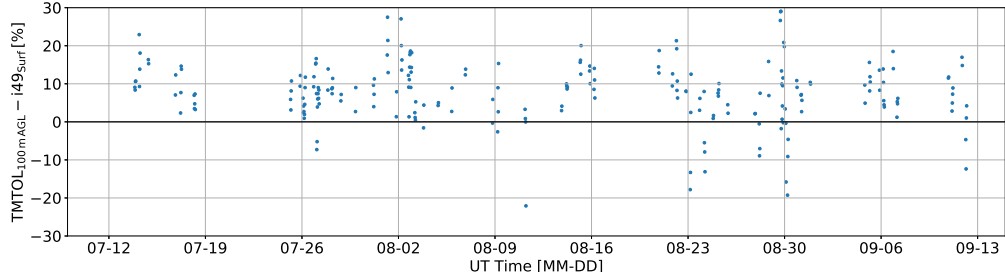

**Figure 12.** Relative difference between the first valid lidar data point (100 m AGL) and the collocated i49 surface ozone meter for the period comprehended between 13 July and 12 September 2018. Each point corresponds to a 30 minute average.

Since the start of the autonomous operations to September 12th 2018, 139 1-hour measurement periods during TROPOMI overpasses were acquired. This extensive dataset not only provides valuable information for the analysis and evaluation of TROPOMI retrievals, but also demonstrates the potential of autonomous operations for the validation of the upcoming TEMPO mission. Additionally, based on this experience, lidar hardware and automation algorithms developed as part of TMTOL
automation upgrade are expected to be implemented on the other TMF lidars, allowing schedule-based autonomous operations for all three TMF lidars.

In order to overcome the issues associated with the tethered balloon operations at TMF, there are plans underway to deploy an ozone monitor on an unmanned aerial vehicle to provide further validation data for the very-near-range receiver as well as to investigate the lowermost 70 m AGL, not covered by the new receiver.

Finally, another upgrade to include aerosol measurement capability is currently being developed. Such upgrade would allow to better characterize the distribution and properties of the aerosols typically found over TMF and thus, better determine to which extent this might affect the ozone retrievals.

*Data availability.* Part of the lidar data used for this study is publicly available at TOLNet (https://www-air.larc.nasa.gov/missions/TOLNet/) and NDACC (http://www.ndacc.org/) websites. For additional data or information please contact the authors.

*Competing interests.* The authors declare that they have no conflict of interest.

*Acknowledgements.* The work described here was carried out at the Jet Propulsion Laboratory, California Institute of Technology, under agreements with the National Aeronautics and Space Administration. The authors acknowledge funding from the Tropospheric Chemistry Program of the NASA Earth Science and Division. The authors would like to thank Andy Langford for the fruitful discussion on the stratospheric intrusion event presented in this work. The authors further thank the MERRA-2 team for providing the data used in this study.





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
