# Peer review of "Upgrade and automation of the JPL Table Mountain Facility tropospheric ozone lidar (TMTOL) for near-ground ozone profiling and satellite validation"

_Atmospheric Measurement Techniques, 2018_

## Referee Comment (RC1) · Anonymous Referee #1 · 29 Nov 2018

The manuscript by Chouza et al. describes autonomous technique for an ozone lidar system, introduces a recently added near-surface channel, and presents intercomparison results of validation using other in situ measurements. This manuscript is well written in general, is technically sound, and has some original aspects which could be valuable for other lidar groups. I have two slightly major concerns/comments: 1) clarify the 5% accuracy in the abstract to avoid confusion and 2) suggest adding description of the auto alignment feature. I recommend to publish after addressing these comments.

Specific comments:

[Figure]

P1, L8-9, "An agreement of 5 % or better with the ozonesonde down to an altitude range of 100-m above ground was demonstrated." It looks very ambitious to conclude the accuracy "5%" just based on one intercomparison experiment between the lidar and tethered ozonesonde (Figure 8). There are two coincident ozonesonde profiles in the May 25 case study (page 14-18 and Figure 11). The discrepancies between the sondes and lidar are not quantified well in the context although the explanations about these discrepancies are provided. But it can be seen that these discrepancies have exceeded 5% at many altitudes even excluding the STE layers in Figure 11 by a visual check. So, it looks like the authors made favorable conclusions by using selective results. Please clarify.

P1, L 13-14, "These comparisons revealed localized differences between sonde and lidar, possibly owing to the differing vertical resolutions (about 52m for lidar and at least 120m for sonde)." What is the corresponding altitude range for the 52-m resolution for the lidar retrievals? 52m is probably the highest resolution of the lidar while ozonesonde has a constant resolution about 100m from surface to $\sim$35km.

P2, L9, have you given what TEMPO stands for somewhere?

P3, L16-17, pure H2 or D2 without buffer gas? Can you give the conversion efficiencies (or the UV pulse energy sent into the sky) for both wavelengths?

P3, L26, what are the sizes of the receivers for the low altitude channel?

P4, L2, what does APC stand for? There are a lot of acronyms in this paper. They are well known in some particular fields (e.g., computer science, satellite), but probably not so for everyone and need to spell out at the 1st time in use.

P5, L13, "the alignment module was. . . the acquisition." Alignment of laser beam with receiver is critical for correct lidar retrievals. Since autonomous alignment is an important feature of the whole autonomous system, can you provide additional description about how your auto alignment works?

P6, L1, Table 1 showing detailed numbers of lidar profiles in each month is not closely related to this paper.

P7, L10-13, 1) the location of the citation is not appropriate. Suggest to put (Kovalev and Bristow, 1996) and Wang et al., 1997) after "dual DIAL measurements". 2) The statement is a little confusing, or incomplete. Reduction of aerosol interference will naturally reduce the uncertainty in DIAL retrieval. But you say reduction of aerosol interference will increase the ozone retrieval uncertainty potentially by inducing other larger errors. The readers will wonder what are the uncertainty sources with larger errors. Please clarify.

P7-8, the authors try to justify the ozone DIAL retrieval with a wavelength pair of 266-289nm without an aerosol correction with two pages of words. This might be ok by considering that the focus of this paper is the description of the autonomous features of lidar system and this particular lidar is located at a high-elevation site where the sky is relatively clean. But I have following comments: 1) the ozone DIAL can't measure accurately in the presence of fire smoke where the aerosol gradients could be even larger than the assumed profile in Figure 4. Especially forest fire occurs frequently in CA recently. 2) The aerosol layer is assumed at a low altitude, 500m AGL, in Figure 4. The percentage uncertainty due to aerosol interference could be much larger at a higher altitude because of ozone number density typically decreases with altitude in troposphere. In addition, the 266-nm laser is not able to cover the altitude higher than 3000m AGL as you mentioned in the paper. 3) There exist some aerosol correction methods for two-wavelength ozone DIALs such as Alvarez II et al. 2011 (cited earlier for another purpose); Immler 2003, Kuang et al., 2011, Sullivan et al., 2014. I believe you still need to mention them although you don't apply aerosol correction because of some reasons.

P10, in terms of the error induced by the timing offset between the two wavelengths, similar issue has been investigated in previous literature (e.g., Figure 4 in Kuang et al., 2013). How does your result compare to previous ones?

P12, L24, "there is a very good agreement (less than 5% difference)...". This statement is not accurate since you mention one sentence later "some deviations between 70 m AGL and 100 AGL" are "less than 10%". Maybe should change to say "mostly less than 5%" or constrain the altitudes above 100m.

P12, L29, "a 17-sample Savitsky-Golay derivative filter was used". Why is the filtering window length different with "9 samples for the analog detection mode and 31 samples for the photon-counting mode" which was mentioned earlier?

P18, L21, "a good correlation between these two quantities is observed", what is the correlation coefficient between the Lidar and i49? Is 10% (a mean value) an exact value or an approximation? Looks like Fig. 12 has sufficient samples so that an exact value of the mean difference can be statistically meaningful.

P18, L25, "The result of this comparison show a very similar difference of about 5-10% between ozone sonde measurements at 100m AGL and surface ozone measurements." If "the result" is an unpublished result and is not given in the paper either, I suggest adding "not shown" to avoid confusion.

P18, L 31, "As a result, the system is now capable of performing measurements autonomously from about 70m AGL up to about 15km." Also P19, L9, "...the lowermost 70m AGL...". There are large discrepancies between the lidar and tethered sonde below 100 m (Figure 8), which actually has not been explained very in the context. Moreover, the first lidar altitude grid is also 100m for the intercomparison with the surface observations (Figure 12). These facts clearly suggest the lidar retrievals below 100m are relatively unreliable. I suggest changing "70m" to "100m" to avoid confusion.

---

## Referee Comment (RC2) · Anonymous Referee #2 · 30 Nov 2018

Chouza et al. gives an overview of the current state of the JPL tropospheric ozone lidar and reports about the automatization efforts as well as the implementation of a new very-near range receiver. The quality of the measurements are validated with other measurements and the long-term stability of the very-near range receiver is shown. Overall the manuscript is well written and gives reasonable explanation in regard to the technical developments.

General comments:

- In section 2 "Instrument description", the reader would benefit from a diagram of the transmitter and receiver layout of the lidar to better follow the instrument description.

- Through the manuscript, some acronyms are frequently used but not explained e.g. TROPOMI or TEMPO whereas other acronyms which are used only once are explained e.g. SRS.

- The authors use AGL and ASL in their altitude nomenclature. This might be confusing to the reader especially if the figure shows the ASL altitude, but the text is describing it in AGL altitude. The figures would need both labels and more consistency would improve the manuscript' Ideally, only one of the two is used.

- Even though "satellite validation" is stated in the title, an example of at least one simultaneous measurement of the lidar and satellite is missing.

Specific comments:

P3L15: SRS is only used once, no need to add the acronym.

P3L24-25: FOV is only used once, no need to add the acronym. It is not clear how two fibers in the focal point of the telescope can have two FOVs and separate the atmospheric volume. Also, what are the size of the FOVs.

Table 1: Clarify if the observations were simultaneously and in the same volume with TROPOMI. If so, it would be very interesting to see at least one of the measurements.

P6L5: ABL is only used once, no need to add the acronym.

P7L16-19: This is a very long sentence and cloud be separated.

P7L22-23: Consider removing the indent and combine the two paragraphs since the second paragraph refers to the first one and gives another argument why this technique can't be used.

P8L14-16: It is not clear why a pellicle beamsplitter is preferred since the other detector

can be adjusted to have no differences in the overlap function. Clarify "overlap function" because the same term is used in P10L12.

P10L29-31: With the information of the typical concentration Figure 6 becomes meaningful. Consider adding it to the figure description or change the label to Error percentage.

P11L8-9: When conducting this test, I wonder how the different wavelengths dependence of the PMT from 288.9nm to 260nm influence the result.

P12L17: Clarify if the measurements are co-located / same volume measurements.

Figure 7: The Error becomes only meaningful if related to a value / expressed in percentage.

Figure 8. Indicate that the ASL altitude is used.

P14L7-8: In this sentence, the ASL and AGL altitude is used. A consistent use of one or the other would help the reader.

P15L4 and Figure 10: Please clarify what surface ozone measurements unit is used.

P16L1-2: Adding a subtraction plot would clearly show the good agreement.

P18L1-2: Figure 10 is in ASL altitude, but the text uses AGL altitude.

P18L3: Please clarify the sentence "As can be seen in Fig. 11b".

P18L17: The time frame of the "Long-term stability" test is very short. Typically, with the change of the yearly seasons (e.g. temperature, humidity) the stability of an optical system might change. I would consider this test as a preliminary "Long-term stability" test.

P19L5: It is referred to other TMF lidars, what are they.

Technical comments:

СЗ

Please check through the manuscript if "collocated" or "co-located" is the right term.

P5L22: replace "S5P" with "ESA Sentinel-5 Precursor"

P7L5: add .... "of" the Raman cells...

P7L17: add .... "of" satellite-borne....

P10L7: "the the" should read "that the"

P10L8: remove .... "on" the receiver

P10L19: "use" should read "used

---

## Author Comment (AC1) · 11 Jan 2019

The comment was uploaded in the form of a supplement:
https://www.atmos-meas-tech-discuss.net/amt-2018-337/amt-2018-337-AC1-supplement.zip

---

## Author Response (AR1)

Dear reviewers,

We like to thank you for your helpful comments on our paper titled "Upgrade and automation of the JPL Table Mountain Facility tropospheric ozone lidar (TMTOL) for near-ground ozone profiling and satellite validation".

The original comments are in bold, followed by our replies. An additional file, highlighting the changes introduced in the manuscript, is also included.

**Reviewer #1**

**General comments:**

**1) clarify the 5% accuracy in the abstract to avoid confusion**

Based on your comments, some modifications were introduced in the abstract (see first specific comment).

**2) suggest adding description of the auto alignment feature.**

Based on your suggestion, the following explanation was included as part of the system automation description:

*The alignment module was implemented to control the alignment of the beams in order to optimize the data quality of the acquisition. For the alignment of each transmitted beam, a step-stare scanning is conducted along two perpendicular axes. After finishing the scanning along the first axis, the centroid of the acquired data as function of the mirror position is calculated at a given alignment altitude and the mirror readjusted to the calculated position. This procedure is then repeted for the second axis. Due to the narrower FOV of the high altitude receiver compared to the other two receiver sets, transmitted beams are aligned using the signals from the high altitude channels at about 5000 m AGL. The aligment of the other two receivers is manually checked about two times a year, although no large readjustments are typically required.*

**Specific comments:**

**P1, L8-9, "An agreement of 5 % or better with the ozonesonde down to an altitude range of 100-m above ground was demonstrated." It looks very ambitious to conclude the accuracy "5%" just based on one intercomparison experiment between the lidar and tethered ozonesonde (Figure 8). There are two coincident ozonesonde profiles in the May 25 case study (page 14-18 and Figure 11). The discrepancies between the sondes and lidar are not quantified well in the context although the explanations about these discrepancies are provided. But it can be seen that these discrepancies have exceeded 5% at many altitudes even excluding the STE layers in Figure 11 by a visual check. So, it looks like the authors made favorable conclusions by using selective results. Please clarify.**

Comparing a lidar retrieval with a free-flying ozonesonde is always complex, as the measured volume might be quite different. This is especially true in cases like the ones shown as part of the 24-h run, when wind speeds were between 15 and 20 m/s. Additionally, the long-term comparison shown in Fig. 13 indicate an agreement better that 10%, even considering the 100 m difference between the ozone photometer.

Having said this, we included some rewording in the abstract to take in account the fact that only one lidar - tethered balloon comparison is shown:

*"An agreement of 5 % or better with the ozonesonde down to an altitude range of 100-m above ground was demonstrated."* was changed to *"An agreement of about 5 \% with the ozonesonde down to an altitude range of 100-m above ground was observed."*

Additionally, results corresponding to the long-term comparison conducted between the surface ozone meter and the first valid retrieved bin are now included:

*"Comparisons between the lowest point retrieved by the lidar and a co-located surface ozone photometer showed no sign of drift over a two-month test period and an agreement better than 10 %."*

**P1, L 13-14, "These comparisons revealed localized differences between sonde and lidar, possibly owing to the differing vertical resolutions (about 52m for lidar and at least 120m for sonde)." What is the corresponding altitude range for the 52-m resolution for the lidar retrievals? 52m is probably the highest resolution of the lidar while ozonesonde has a constant resolution about 100m from surface to ~35km.**

The 52-m resolution corresponds to the lowest part of the retrieval (analog signal from the new very-near-range receiver). In order to clarify the statement, the following change was made:

*"These comparisons revealed localized differences between sonde and lidar, possibly owing to the differing vertical resolutions (between 52 m and 380 m for lidar and about 100m for the sonde)."*

Further details of the lidar resolution scheme used for the lidar/sonde comparison can be found in P15, L6-9.

**P2, L9, have you given what TEMPO stands for somewhere?**

An explanation of the TEMPO (Tropospheric Emissions: Monitoring Pollution) acronym is now included in the text.

**P3, L16-17, pure H2 or D2 without buffer gas? Can you give the conversion efficiencies (or the UV pulse energy sent into the sky) for both wavelengths?**

No buffer gas is used in the Raman cells. Unfortunately no accurate conversion efficiencies can be provided at this point. Reference test conducted on this topic at TMF can be found in Haner et al., 90, nevertheless the cell length is now different (4 m vs 2 m in the cited paper).

**P3, L26, what are the sizes of the receivers for the low altitude channel?**

The diameter of the low altitude receivers (50 mm) is now included as part of the description.

**P4, L2, what does APC stand for? There are a lot of acronyms in this paper. They are well known in some particular fields (e.g., computer science, satellite), but probably not so for everyone and need to spell out at the 1st time in use.**

Based on the reviewer recommendations, explanations for the following acronyms were included: TROPOspheric Monitoring Instrument (TROPOMI), Tropospheric Emissions: Monitoring

Pollution (TEMPO), differential absorption lidar (DIAL), Hierarchical Data Format version 5 (HDF5), electrochemical concentration cell (ECC).

**P5, L13, "the alignment module was. . . the acquisition." Alignment of laser beam with receiver is critical for correct lidar retrievals. Since autonomous alignment is an important feature of the whole autonomous system, can you provide additional description about how your auto alignment works?**

Additional description of the alignment procedure is now included (answered in the general comments section).

**P6, L1, Table 1 showing detailed numbers of lidar profiles in each month is not closely related to this paper.**

Table 1 is included to provide an insight of the datasets already available for TROPOMI validation.

**P7, L10-13, 1) the location of the citation is not appropriate. Suggest to put (Kovalev and Bristow, 1996) and Wang et al., 1997) after "dual DIAL measurements". 2) The statement is a little confusing, or incomplete. Reduction of aerosol interference will naturally reduce the uncertainty in DIAL retrieval. But you say reduction of aerosol interference will increase the ozone retrieval uncertainty potentially by inducing other larger errors. The readers will wonder what are the uncertainty sources with larger errors. Please clarify.**

The trade-off between aerosol error correction and random error increase is explained in Wang et al. 97.:

"Therefore, although the three-wavelength dual DIAL appears to have a somewhat larger statistical error, it can be used to reduce substantially the systematic error of the stratospheric ozone measurements in the presence of volcanic aerosols. The system error is also almost unaffected by aerosol loading spatial inhomogeneity and aerosol optical properties."

The sentence was modified in order to better reflect this compromise:

*"Additionally, it has to be mentioned that although dual DIAL measurements help to reduce the systematic error caused by the aerosol influence on the retrieval, it is at the expense of increasing the overall statistical uncertainty on the ozone retrieval (Wang et al., 97)."*

**P7-8, the authors try to justify the ozone DIAL retrieval with a wavelength pair of 266- 289nm without an aerosol correction with two pages of words. This might be ok by considering that the focus of this paper is the description of the autonomous features of lidar system and this particular lidar is located at a high-elevation site where the sky is relatively clean. But I have following comments: 1) the ozone DIAL can't measure accurately in the presence of fire smoke where the aerosol gradients could be even larger than the assumed profile in Figure 4. Especially forest fire occurs frequently in CA recently. 2) The aerosol layer is assumed at a low altitude, 500m AGL, in Figure 4. The percentage uncertainty due to aerosol interference could be much larger at a higher altitude because of ozone number density typically decreases with altitude in troposphere. In addition, the 266-nm laser is not able to cover the altitude higher than 3000m AGL as you mentioned in the paper. 3) There exist some aerosol correction methods for two-wavelength ozone DIALs such as Alvarez II et al. 2011 (cited earlier for another purpose); Immler 2003, Kuang et al., 2011, Sullivan et al., 2014. I believe you still need to mention them although you don't apply aerosol correction because of some reasons.**

We agree with the comments made by the reviewer regarding the limited range of the selected wavelength pair and the need of aerosol correction methods in order to mitigate the effects of strong aerosol plumes in the retrieval accuracy. Nevertheless, and as the reviewer also points out, the main purpose of the wavelength selection discussion is to justify the election of the wavelength DIAL pair for the new very-near-range receiver considering that only a range of about 1000 m is required, that the aerosol load at TMF is typically small, and that there are only a few possible combinations based on the current TMTOL setup. The wavelength selection discussion doesn't intend to provide a general answer to the DIAL wavelength selection problem, but a solution for a specific problem under specific conditions.

Since ozone studies in the presence of different types of aerosols might be part of the applications of the new very-near-range receiver, the following sentence was added:

*In case of stronger aerosol loads, the aerosol influence might be further reduced by applying an aerosol correction algorithm (Alvarez et al., 2011; Immler, 2003; Kuang et al., 2011; Sullivan et al., 2014).*

**P10, in terms of the error induced by the timing offset between the two wavelengths, similar issue has been investigated in previous literature (e.g., Figure 4 in Kuang et al., 2013). How does your result compare to previous ones?**

Results are similar after considering the difference in the DIAL wavelengths used in both cases. A citation for the mentioned article was included.

**P12, L24, "there is a very good agreement (less than 5% difference). . .". This statement is not accurate since you mention one sentence later "some deviations between 70 m AGL and 100 AGL" are "less than 10%". Maybe should change to say "mostly less than 5%" or constrain the altitudes above 100m.**

The suggested addition ("*mostly less than 5%*") was included.

**P12, L29, "a 17-sample Savitsky-Golay derivative filter was used". Why is the filtering window length different with "9 samples for the analog detection mode and 31 samples for the photon-counting mode" which was mentioned earlier?**

The difference in the window length is related to the purpose of the analysis. In the case of Figure 7, the analysis focuses on the systematic errors introduced by the differential efficiency component and its relation with the minimum achievable range. A short window allows to get a better spatial resolution on the near end of the retrieval at the price of increasing the random uncertainty. In the case of comparisons with radiosondes, the window length is increased in order to reduce the random uncertainty. As a consequence, biased points at the lower end of the retrieval are now included in the first bins of the retrieval and the minimum (unbiased) achievable range increases from about 70 m to about 100 m.

**P18, L21, "a good correlation between these two quantities is observed", what is the correlation coefficient between the Lidar and i49? Is 10% (a mean value) an exact value or an approximation? Looks like Fig. 12 has sufficient samples so that an exact value of the mean difference can be statistically meaningful.**

Mean difference and number of points is now included as part of Fig. 13 (previously Fig. 12). The text was modified to reflect the accurate number (7.2% instead of 10%).

**P18, L25, "The result of this comparison show a very similar difference of about 5- 10% between ozone sonde measurements at 100m AGL and surface ozone measurements." If "the result" is an unpublished result and is not given in the paper either, I suggest adding "not shown" to avoid confusion.**

"Not shown" was included at the end of the sentence to avoid confusion.

**P18, L 31, "As a result, the system is now capable of performing measurements autonomously from about 70m AGL up to about 15km." Also P19, L9, ". . .the lowermost 70m AGL. . .". There are large discrepancies between the lidar and tethered sonde below 100 m (Figure 8), which actually has not been explained very in the context. Moreover, the first lidar altitude grid is also 100m for the intercomparison with the surface observations (Figure 12). These facts clearly suggest the lidar retrievals below 100m are relatively unreliable. I suggest changing "70m" to "100m" to avoid confusion.**

Minimum altitude was changed to 100 m following the observations of the reviewer.

**Reviewer #2**

**General comments:**

**- In section 2 "Instrument description", the reader would benefit from a diagram of the transmitter and receiver layout of the lidar to better follow the instrument description.**

Following the reviewer suggestion, a diagram of the TMTOL was included (Fig. 1).

**- Through the manuscript, some acronyms are frequently used but not explained e.g. TROPOMI or TEMPO whereas other acronyms which are used only once are explained e.g. SRS.**

A general revision of the acronym use was conducted.

**- The authors use AGL and ASL in their altitude nomenclature. This might be confusing to the reader especially if the figure shows the ASL altitude, but the text is describing it in AGL altitude. The figures would need both labels and more consistency would improve the manuscript' Ideally, only one of the two is used.**

We partially agree with the observations of the reviewer with regard to the use AGL and ASL. Nevertheless, since Table Mountain Facility (TMF) is located at a relatively high altitude (2285 m ASL) and part of this work focuses in very-near range measurements (between 100 m and 1000 m AGL), we also consider that some aspects of the work are better explained using AGL altitude while others can be better analyzed using ASL altitude. For example, in the case of the very-near-range receiver technical discussion, differential efficiency and the effects of acquisition delay between both channels of the DIAL system are only relative to the lidar position (very similar to AGL altitude) and independent of its ASL altitude. In the second part of the discussion, when the system performance is compared to tethered-balloon measurements and the stratospheric intrusion event is analyzed, the use of ASL altitude emphasizes the fact that TMF is at a relatively high altitude.

Based on this considerations, some changes were made in the use of ASL and AGL to improve the manuscript consistency.

**- Even though "satellite validation" is stated in the title, an example of at least one simultaneous measurement of the lidar and satellite is missing.**

Unfortunately the TROPOMI ozone profile data product is still under development and no data is yet available for publication. The following sentence was added to clarify this:

*"Unfortunately the TROPOMI ozone profile data product is still under development. For this reason, no comparisons are included in this work."*

**Specific comments:**

**P3L15: SRS is only used once, no need to add the acronym.**

This is addressed in a previous answer to reviewer #1.

**P3L24-25: FOV is only used once, no need to add the acronym. It is not clear how two fibers in the focal point of the telescope can have two FOVs and separate the atmospheric volume. Also, what are the size of the FOVs.**

The acronym is now used also as part of the auto-alignment description. Regarding the wavelength separation, the paper mentioned that both fibers are in the same focal plane, not the same focal point. By keeping fibers in the same focal plane and slightly displacing them from the focal point, two FOVs can be achieved. Further information regarding this setup (i.e. FOVs) can be found in McDermid et al., 2002.

**Table 1: Clarify if the observations were simultaneously and in the same volume with TROPOMI. If so, it would be very interesting to see at least one of the measurements.**

Experiments conducted for TROPOMI validation consist in 1-hour measurement periods centered around the predicted daily overpass time. This is now clarified as part of Table 1 discussion:

*Together with the regular 2-hour experiments typically conducted at TMF as part of NDACC activities, regular 1-hour daytime measurements were carried out during forcasted overpasses in support of the validation of the TROPOspheric Monitoring Instrument (TROPOMI) onboard the Sentinel-5 Precursor satellite.*
The lack of a comparison with TROPOMI ozone profiles is explained in a previous answer.

**P6L5: ABL is only used once, no need to add the acronym.**

The acronym was removed.

**P7L16-19: This is a very long sentence and cloud be separated.**

Sentence was reformulated.

*Because the daytime operation is important for the validation satellite-borne ozone instruments and considering the low intensity of the Raman scattering (about two orders of magnitude lower than Rayleigh scattering) the use of wavelengths in the blind region of the solar spectrum is preferred.*

**P7L22-23: Consider removing the indent and combine the two paragraphs since the second paragraph refers to the first one and gives another argument why this technique can't be used.**

Following the reviewer suggestions, the paragraphs were merged.

**P8L14-16: It is not clear why a pellicle beamsplitter is preferred since the other detector can be adjusted to have no differences in the overlap function. Clarify "overlap function" because the same term is used in P10L12.**

Although the detector position might be adjusted to compensate the optical path difference introduced by a thick beamsplitter, this would add additional complexity to the setup and an additional degree of freedom at the time of aligning the system. Additionally, a pellicle beamsplitter eliminates ghosts, which also can affect the behavior of the system in the near range regime.

**P10L29-31: With the information of the typical concentration Figure 6 becomes meaningful. Consider adding it to the figure description or change the label to Error percentage.**

Typical concentration was included as part of the figure description.

**P11L8-9: When conducting this test, I wonder how the different wavelengths dependence of the PMT from 288.9nm to 260nm influence the result.**

Since the PMT efficiency is not range dependent, no contribution to the differential efficiency is expected.

**P12L17: Clarify if the measurements are co-located / same volume measurements.**

Co-located was added to clarify the type of comparison described in the section.

*"...a set of measurements by a co-located ozonesonde tethered system..."*

**Figure 7: The Error becomes only meaningful if related to a value / expressed in percentage.**

Typical concentration was included as part of the figure description.

**Figure 8. Indicate that the ASL altitude is used.**

Figures 8, 10 and 12 were modified to indicate that ASL altitude is ised.

**P14L7-8: In this sentence, the ASL and AGL altitude is used. A consistent use of one or the other would help the reader.**

Since for some aspects of the discussion (i.e. comparison with previous discussion about differential efficiency). In order to clarify the discussion, ASL altitude is reported together with the AGL altitude.

**P15L4 and Figure 10: Please clarify what surface ozone measurements unit is used.**

In all cases, surface ozone measurements were conducted by a co-located Thermo Scientific Model 49i unit. This is now clarified in the Figure 10 description as well as in the corresponding discussion:

*"...surface ozone measurements carried out by the co-located Thermo Scientific Model 49i."*

**P16L1-2: Adding a subtraction plot would clearly show the good agreement.**

A subtraction plot was added to the figure.

**P18L1-2: Figure 10 is in ASL altitude, but the text uses AGL altitude.**

As in the previous case, ASL altitude is included in the discussion to simplify the reading.

**P18L3: Please clarify the sentence "As can be seen in Fig. 11b".**

Further description of the well-mixed layer is now included as part of the figure description.

*As can be seen in Fig. 12b, this low ozone layer sits on top of a well-mixed layer (no vertical gradient in the potential temperature) that extends up to 120 m AGL (2405 m ASL) and is trapped between two inversions.*

**P18L17: The time frame of the "Long-term stability" test is very short. Typically, with the change of the yearly seasons (e.g. temperature, humidity) the stability of an optical system might change. I would consider this test as a preliminary "Long-term stability" test.**

We agree with the observations of the reviewer. Preliminary was added to the comparison description to emphasize this.

*"...a preliminary long-term comparison between the..."*

**P19L5: It is referred to other TMF lidars, what are they.**

The other TMF lidars are a stratospheric ozone lidar and a water vapor Raman lidar. The following clarification was included:

*...(a stratospheric ozone lidar and a water vapor Raman lidar)...*

**Technical comments:**

**Please check through the manuscript if "collocated" or "co-located" is the right term.**

Corrected.

**P5L22: replace "S5P" with "ESA Sentinel-5 Precursor"**

This is replaced in the new manuscript version.

**P7L5: add . . ."of" the Raman cells. . .**

Corrected.

**P7L17: add . . ."of" satellite-borne. . ..**

Corrected.

**P10L7: "the the" should read "that the"**

Corrected.

**P10L8: remove . . ."on" the receiver**

Corrected.

**P10L19: "user" should read "used"**

Corrected.

[revised manuscript text omitted]

The paper is organized as follows. Section 2 provides a brief description of the system previous to the modifications presented in this paper. Section 3 describes the hardware and software modifications introduced as part of the system automation. Section 4 provides a description of the new receiver for very-near-range measurements. Section 5 presents the validation of the new

35 receiver by means of a comparison with an ozone sensor deployed on a tethered balloon as well as a case study that shows the

[Figure]

**Figure 1.** Current transmitter and receiver layout of TMTOL. BEX stands for beam expander. The new very near-range receiver is also shown.

[revised manuscript text omitted]
. For the alignment of each transmitted beam, a step-stare scanning is conducted along two perpendicular axes. After finishing the scanning along the first axis, the centroid of the acquired data as function of the mirror position is calculated at a given alignment altitude and the mirror readjusted to the calculated position. This procedure is then repeted for the second axis. Due to the narrower FOV of the high altitude receiver compared to the other two receiver sets, transmitted

15   beams are aligned using the signals from the high altitude channels at about 5000 m AGL. The aligment of the other two receivers is manually checked about two times a year, although no large readjustments are typically required. Finally, the acquisition module is in charge of retrieving the data from the Licel transient recorders and storing the data.

[revised manuscript text omitted]

As can be seen, for the delay observed between the two transient recorders ($23\,\mathrm{ns}$ or $3.45\,\mathrm{m}$), an overestimation of about $0.5 \times 10^{18}\,\mathrm{m^{-3}}$ is introduced. For a typical concentration of $1 \times 10^{18}\,\mathrm{m^{-3}}$, this represents an overestimation in the ozone concentration of about 50 percent at $100\,\mathrm{m}$. By introducing a relative delay between the trigger signals of both transient recorders, this error source was eliminated.

After correcting the effects of the relative trigger delay, and in order to further investigate the systematic error sources of the receiver, an estimation of the differential efficiency was performed. The differential efficiency condenses a set of different error

[Figure]

**Figure 7.** Error generated as a function of the delay $\Delta R$ between the $266\,\mathrm{nm}$ and the $288.9\,\mathrm{nm}$ channels. Positive delays correspond the the $266\,\mathrm{nm}$ channel being triggered after the $288.9\,\mathrm{nm}$ channel. A typical ozone concentration is about $1 \times 10^{18}\,\mathrm{m}^{-3}$.

[revised manuscript text omitted]
 ($2385\,\mathrm{m}$ ASL), together with $1-\sigma$ uncertainty. d) difference between the lidar retrieval at $100\,\mathrm{m}$ AGL ($2385\,\mathrm{m}$ ASL) and the surface ozone meter (i49).

receivers ($380\,\mathrm{m}$ vertical resolution). In the overlap region between the very-near-range and the mid-range receiver, the mean is presented. The temporal resolution is 30 minutes.

A first qualitative comparison indicate a good agreement between the top of the very-near-channel and the bottom of the mid-range channels as well as between the $100\,\mathrm{m}$ AGL ($2385\,\mathrm{m}$ ASL) lidar retrieval and the surface ozone measurements

carried out by the co-located Thermo Scientific Model 49i. Although this doesn't provide a full validation of the new channel, it provides further confidence to the tests performed with the tethered balloon system and presented in the previous subsection.

The high ozone concentration visible in Fig. 11a above $3000\,\mathrm{m}$ ASL is in qualitative agreement with a stratospheric intrusion event and with the forecast/reanalysis provided by the models. Below the intrusion, a thin layer of about $200\,\mathrm{m}$ depth characterized by a relatively low ozone concentration can also be recognized. Since this kind of fine structures are typically not reproduced by operational models and since they provide valuable information to evaluate the performance of the receiver, a comparison between the lidar retrieval and the two ozonesondes launched during the experiment is presented in Fig. 12. This will allow to determine whether or not this corresponds to a real feature and what is the expected accuracy of the new receiver.

Figure 12 presents the intercomparison between the two ozone sondes and the lidar-retrieved ozone profiles. Potential temperature, humidity and wind profiles retrieved by the radiosonde are also shown. During the launch of this first sonde (4:40 UT), no strong signature of the atmospheric intrusion was visible neither in the lidar nor in the ozone sonde. A general good agreement between the lidar- and the sonde-retrieved ozone profile is visible in Fig. 12a. In the very-near range, the thin layer of low ozone concentration visible in Fig. 11 can be found between $100\,\mathrm{m}$ and $200\,\mathrm{m}$ AGL (2385 m and 2485 m ASL), while in the case of the sonde, a thicker and less pronounced layer of low ozone concentration was observed between $200\,\mathrm{m}$ and $700\,\mathrm{m}$ AGL (2485 m and 2985 m ASL). As can be seen in Fig. 12b, this low ozone layer sits on top of a well mixed layer (no vertical gradient in the potential temperature) that extends up to $120\,\mathrm{m}$ AGL (2405 m ASL) 
[revised manuscript text omitted]

Immler, F.: A new algorithm for simultaneous ozone and aerosol retrieval from tropospheric DIAL measurements, Applied Physics B, 76, 593–596, 2003.

Kovalev, V. A. and Bristow, M. P.: Compensational three-wavelength differential-absorption lidar technique for reducing the influence of differential scattering on ozone-concentration measurements, Applied optics, 35, 4790–4797, 1996.

Kuang, S., Newchurch, M., Burris, J., Wang, L., Buckley, P. I., Johnson, S., Knupp, K., Huang, G., Phillips, D., and Cantrell, W.: Nocturnal ozone enhancement in the lower troposphere observed by lidar, Atmospheric environment, 45, 6078–6084, 2011.

[revised manuscript text omitted]